# Modeling and Analysis of Wave Energy Harvester with Symmetrically Distributed Galfenol Cantilever Beams

**DOI:** 10.3390/ma16165585

**Published:** 2023-08-11

**Authors:** Sunyangyang Jin, Aihua Meng, Mingfan Li, Zhenlong Xu, Shuaibing Wu, Yu Chen

**Affiliations:** 1School of Mechanical Engineering, Hangzhou Dianzi University, Hangzhou 310018, China; 2School of Mechanical and Automotive Engineering, Zhejiang University of Water Resources and Electric Power, Hangzhou 310018, China

**Keywords:** wave energy, Galfenol, energy harvester, cantilever

## Abstract

In response to the challenges of difficult energy supply and high costs in ocean wireless sensor networks, as well as the limited working cycle of chemical batteries, a cylindrical wave energy harvester with symmetrically distributed multi-cantilever beams was designed with Galfenol sheet as the core component. The dynamic equation of the device was established, and ANSYS transient dynamic simulations and Jiles-Atherton hysteresis model analysis were conducted to develop a mathematical model of the induced electromotive force of the Galfenol cantilever beam as a function of deformation. Experimental validation demonstrated that the simulated results of the cantilever beam deformation had an average error of less than 7% compared to the experimental results, while the average error between the theoretical and experimental values of the induced electromotive force of the device was around 15%, which preliminarily verifies the validity of the mathematical model of the device, and should be subject to further research and improvement.

## 1. Introduction

In order to better develop and protect the marine environment, marine environmental monitoring technology plays a crucial role [1,2]. Currently, marine environmental monitoring technology mainly relies on wireless sensor networks, but with the development and application of marine wireless sensor network technology, the difficulty in energy supply has become the main problem limiting its development.

As wireless sensor networks have the advantages of wide distribution and low energy consumption, traditional large-scale wave energy conversion devices [3,4,5] are not suitable for their power supply. Currently, existing small-scale marine wave energy harvesting technologies mainly include electromagnetic, piezoelectric, frictional, and magnetostrictive methods. Shao [6] developed a multifunctional mixed wave energy harvester based on nanoscale friction and electromagnetic technology, while Masoumi developed an electromagnetic wave energy converter that utilizes magnetic levitation [7].

For electromagnetic wave energy harvesters, power is generated by cutting magnetic flux lines through a coil, which has the advantage of not requiring a driving power supply. However, it has the disadvantages of poor environmental adaptability, serious magnetic leakage, and large volume. Viet developed a wave energy harvester based on the piezoelectric effect [8], which has the advantages of being small in size and having a simple structure. However, due to the large internal resistance of the device, its power output is relatively low.

Wang has successfully developed a new energy technology called the “triboelectric nanogenerator” [9], which uses the coupling of frictional charging and electrostatic induction to generate electricity. However, the low durability and reliability of the frictional materials result in short service life and limit its development. 

Liu invented a wave energy harvester based on columnar magnetostrictive materials [10], which uses the Villari effect of magnetostrictive materials and Faraday’s law of electromagnetic induction to generate electricity. 

For magnetostrictive wave energy harvesting devices, magnetostrictive materials are ideal materials for energy conversion due to their excellent mechanical properties, magnetomotive coupling coefficients that are two to three times that of piezoelectric ceramic materials, the absence of depolarization issues, and high energy conversion efficiency [11]. In addition, giant magnetostrictive materials have a wide frequency range and are suitable for low-frequency environments, such as wave energy harvesting. If they are applied to energy harvesting in ocean waves with large spectral variations, they could have considerable development potential and application prospects. Based on the working mechanism of magnetostrictive materials and the application scenarios of wave energy harvesting devices, this paper designs a wave energy harvesting device and studies its working characteristics.

## 2. Wave Energy Harvester Design

### 2.1. Design of Energy Harvesting Methods

The energy harvesting mechanism is a crucial component of a wave energy harvester, which utilizes the Villari effect of magnetostrictive [12] materials and the electromagnetic induction law to convert the oscillation energy of waves into electrical energy. The design of the structure directly affects the stability and energy-harvesting efficiency of the entire device. Currently, typical magnetostrictive energy harvesting structures mainly use rod and cantilever beam structures [13,14]. The rod structure can harvest vibrational energy at any frequency and can be made into a closed magnetic circuit to avoid magnetic leakage and effectively improve the electromechanical conversion efficiency of the material. However, when applied to wave energy harvesting structures, it is more complex, larger in size, difficult to apply stress, and not easy to harvest electrical energy. The cantilever beam [15] structure is relatively simple, smaller in size, and has a wide range of applications. It is easy to harvest force deformation, but for efficiency reasons, cantilever beams need to operate close to the resonant frequency. Moreover, Galfenol material has better mechanical properties than Terfenol-D [16], a higher electromechanical coupling coefficient, and lower production costs. Comprehensive analyses have shown that the magnetostrictive material Galfenol is more suitable for energy harvesting in the variable and unstable marine environment, and the structural form adopts a cantilever beam structure.

Based on the working scenario and vibration energy conversion mechanism of wave energy harvesters, the typical application method for these devices is to float on the water surface. However, due to the uncertainty of the wave direction, a multi-cantilever beam annular energy harvesting mechanism has been designed, as shown in Figure 1, to collect wave energy from multiple directions on the water surface. This mechanism features a circular base and four cantilever beams evenly distributed around the circumference, forming a symmetrical structure that improves overall stability and enhances the device’s power generation capabilities.

### 2.2. Overall Structural Design

Since the waves on the sea’s surface have large fluctuations, it is conducive to harvesting wave energy. Therefore, a floating mechanism is designed to provide buoyancy for the device to float on the water’s surface. To prevent the device from capsizing and detaching from the water’s surface under the action of waves, a gravity anchor mechanism is designed to ensure the stability of the device. Based on these requirements, the wave energy harvester shown in Figure 2 was designed, with the exploded view on the left and the external structure on the right. The device is small in size, simple in structure, and all components are mainly cylindrical, which is conducive to the overall structural balance and stability.

Under the action of waves, the wave energy harvester floats up and down on the ocean surface, resulting in the vibration of the Galfenol sheets. On the basis of the Villari effect, the deformation of the Galfenol film induces a deflection of the magnetic moment of the magnetic domains inside the Galfenol sheet, then changes the magnetization intensity of it. According to Faraday’s law, a changing magnetic field produces an induced electromotive force in the coil wound around the Galfenol thin film. The mechanical energy of waves is converted into electrical energy with the Galfenol energy harvester. The working principle of the Galfenol energy harvester is shown in Figure 3.

## 3. Establishment of Mathematical Model for Wave Energy Harvester

Due to the fact that the excitation signal of the Galfenol cantilever beam’s fixed end base is related to the vibration characteristics of the energy harvester on the water surface, it is necessary to analyze the vibration characteristics of the device under wave excitation. Under the action of water surface waves, the energy harvester will move along with the waves, and since the waveform of the water surface undulation is approximately sinusoidal, the motion of the device can be approximated as a simple harmonic motion [17,18], as shown in Figure 4. Under the action of a sinusoidal wave, the energy harvester will move in three directions: X, Y, and Z. However, it is mainly the motion in the Z-direction that causes the energy-harvesting device to vibrate up and down. Relative to the motion amplitude of the Z direction, the motion of the cantilever beam in the X and Y directions can be ignored.

### 3.1. Dynamical Analysis of the Wave Energy Harvester

To analyze the motion characteristics of the wave energy harvester, it is first necessary to determine the excitation rule of the sinusoidal wave. Taking the sea conditions near Chu Island, Weihai, as the analysis object, the National Centre for Ocean Technology in 2015 analyzed the characteristic regularities of the sea conditions in that area and established a typical wave equation based on the minimum wave height and sea conditions in that area [19], as shown in Table 1. Based on the elements of the sea conditions in that area and the applicability scope of linear wave theory [20], a typical wave surface equation was established using linear wave theory, as shown in Equation (1):(1)$$\begin{array}{l} z_{W}=\frac{H}{2}\cos\left(\mathrm{kx}-\omega t\right)\\ \;\;=0.175\cos\left(0.28x-1.66t\right) \end{array}$$
where $z_{W}$ is the displacement of the wave in the vertical direction, k is the wave number, H is the wave height, and ω is the frequency.

Based on the wave equation established above, a dynamic analysis is conducted on the energy harvester floating on the water’s surface. The device mainly experiences the effects of gravity (G), buoyancy ($F_{S}$), and wave force ($F_{W}$) when it undergoes harmonic motion under the action of waves. The force diagram of the device is shown in Figure 5.

According to Newton’s second law, the motion equation of a floating wave energy harvester can be expressed as follows:(2)$$M\ddot{z}=F_{S}+F_{W}-G$$
where M is the mass of the wave energy harvester, z is the vertical displacement of the device, and the net elastic restoring force $F_{b}$ can be expressed as the difference between buoyancy and gravity, as shown in Equation (2).
(3)$$F_{b}=F_{S}-G=k(z_{W}-z)$$

From Equation (3), k=ρgA, parameter ρ represents the density of seawater, parameter g represents the gravitational constant, parameter A represents the cross-sectional area of the wave energy harvester in contact with the surface of the water and can be calculated as $A=\pi \cdot r^{2}$.

As the wave energy harvester is a small component, its wave force $F_{W}$ can be calculated using the Morison equation. According to the linearized Morison equation [21], the wave force $F_{W}$ is expressed as:(4)$$F_{W}=-\mu \ddot{z}-\lambda \dot{z}$$

From Equation (4), $\mu =\frac{1}{2}\rho C_{D}\sqrt{8/\mu_{rms}}$, $\lambda =C_{M}\rho \pi D^{2}/4$, parameter $C_{M}$ represents the coefficient of inertia and can be set to $C_{M}=2.0$. Parameter D represents the diameter of the structure, and $\mu_{rms}$ represents the root-mean-square velocity.

By combining Equations (2)–(4) with the force analysis of the wave energy harvester, the motion equation of the device can be obtained.
(5)$$\left(M+\mu \right)\ddot{z}+\lambda \dot{z}+kz=kz_{W}$$

It can be seen from Equation (5) that the vertical displacement of the wave energy harvester is mainly affected by its own weight. By substituting the wave Equation (1) into Equation (5), and solving the resulting second-order nonhomogeneous differential equation, the corresponding vertical motion trajectory of the wave energy harvester can be obtained, as shown in the displacement curve in Figure 6.

Because the total mass of the wave energy harvester based on Galfenol is relatively small and it floats on the water surface, it can move with waves. As seen in Figure 6, the motion amplitude and period of the wave energy harvester are similar to the wave characteristics, showing good motion consistency.

### 3.2. Modeling and Simulation of Galfenol Cantilever Beams

Xiong and the Naval Air Systems Command of the United States indirectly studied and verified the power generation performance of wave energy harvesters by using a sine mechanism to mimic the sine excitation of wave loads [22,23]. Therefore, the analysis of the motion characteristics of wave energy harvesters can be approximated and driven by a sinusoidal oscillator. The energy harvester adopts a circumferential symmetrical arrangement structure. Under the action of waves, the four cantilever beams in the energy harvester have the same motion state, so the energy harvester can be simplified to a single cantilever beam in the finite element analysis, ignoring the rest of the structure of the energy harvesting device. 

To study the motion state of Galfenol cantilever beams under displacement excitation, ANSYS workbench’s Transient Structural module is used to simulate and analyze the Galfenol cantilever beams. The up-and-down motion of the wave causes the shell structure of the energy harvesting device to follow the sinusoidal motion, which then causes the cantilever beams to vibrate up and down. In the FEM analysis, this is translated into an up-and-down sinusoidal sliding of the cantilever beam. To facilitate the application of sinusoidal excitation to the Galfenol cantilever beams in the vertical direction in the simulation environment, a rectangular upright plate is added to the left end of the Galfenol cantilever beam base to allow the Galfenol cantilever beam to slide up and down along the plate. By analyzing the vertical displacement of the Galfenol cantilever beam, the vibration deformation of the Galfenol cantilever beam relative to the fixed end of the base during its motion process can be solved. The schematic diagram of the Galfenol cantilever beam under fixed constraints, the gravity field, and moving deputy loading is shown in Figure 7.

Since the mass of the mass block at the free end of the Galfenol cantilever beam affects its deformation, a dynamic structural simulation was conducted on mass blocks with masses of 2 g, 4 g, and 8 g. The deformation in the vertical direction produced by the vibration of the free end of the cantilever beam is shown in Figure 8.

From Figure 8, it can be seen that, under the same excitation conditions, the maximum deformation of the Galfenol cantilever beam slightly increases with the increase in the mass of the mass block at the free end, but the stability of the peak value decreases. By comparing the deformation curves of the three mass blocks, it can be seen that the deformation curve of the Galfenol cantilever beam with a mass block of 2 g has smaller changes between wave peaks, and the deformation variable is relatively stable, with a displacement amplitude of 6.36 mm.

### 3.3. Calculation and Analysis of Induced Electromotive Force of the Wave Energy Harvester

According to Faraday’s electromagnetic induction law, the induced electromotive force produced by the magnetic induction coil wrapped around the Galfenol sheet under the alternating magnetic field is shown in Equation (6).
(6)$$E(t)=N\frac{d\Phi }{dt}=NA\frac{dB}{dt}$$
where N is the number of turns of the magnetic induction coil, Φ is the magnetic flux inside the induction coil, A is the cross-sectional area of the induction coil, and B is the magnetic induction intensity. From the equation of magnetic induction intensity B, we can obtain:(7)$$B=\mu_{0}\left(H+M\right)$$
where $\mu_{0}$ is the magnetic permeability of the vacuum, H is the magnetic field strength, and M is the magnetization intensity. By substituting Equation (7) into Equation (6), when the biased magnetic field H is a constant, the induced electromotive force is given by Equation (8).
(8)$$E(t)=NA\mu_{0}(\frac{dH}{dt}+\frac{dM}{dt})=NA\mu_{0}\frac{dM}{dt}$$

In the absence of external energy losses, the rate of change in magnetization intensity within the Galfenol sheet, denoted as dM/dt, can be expressed as a function of stress, as outlined in Equation (9).
(9)$$\frac{dM}{dt}=\frac{\partial M}{\partial \sigma }\cdot \frac{d\sigma }{dt}$$
where σ is the stress on the Galfenol sheet. By substituting Equation (9) into Equation (8), the induced electromotive force corresponding to the stress changes in magnetization intensity can be obtained, as shown in Equation (10).
(10)$$E(t)=NA\mu_{0}\frac{\partial M}{\partial \sigma }\cdot \frac{d\sigma }{dt}$$

From Equation (10), we can determine the output-induced electromotive force of the wave energy harvester if we know the variation rate of the magnetization intensity M of the Galfenol sheet with respect to stress σ. Based on the Jiles-Atherton hysteresis theory model [24,25], the relationship between the magnetization intensity M of the Galfenol sheet and the stress σ can be established, as shown in Equation (11).
(11)$$M(\sigma ,H)=\frac{M_{S}H}{M_{S}-3a}+c^{\frac{(M_{S}-3a)\sigma^{2}}{2E\xi (3a-cM_{S})}}$$

By combining the expressions for the induced electromotive force, Equation (10) and the total magnetization intensity with respect to stress, Equation (11), the equation for the induced electromotive force with respect to stress can be obtained, as shown in Equation (12).
(12)$$E=NA\mu_{0}\frac{(M_{S}-3a)\sigma M+M_{S}H_{\sigma }}{E\xi (3a-cM_{S})}\frac{d\sigma }{dt}$$

According to the relationship between stress and strain, the deformation of the Galfenol cantilever beam, denoted by z(t), corresponds to a stress, denoted by σ(t), as expressed in Equation (13).
(13)$$\sigma (t)=c^{H}\varepsilon (t)=c^{H}(\frac{z(t)}{l})$$

Finally, by substituting the stress Equation (13) into Equation (12), the induced electromotive force E converted by the wave energy harvester under different deformation displacements can be obtained, as shown in Equation (14).
(14)$$E=NA\mu_{0}c^{H}\frac{(M_{S}-3a)c^{H}\cdot z(t)M+M_{S}H_{\sigma }}{E\xi (3a-cM_{S})l^{2}}\frac{dz(t)}{dt}$$

Using a MATLAB simulation calculation based on Equation (14), the output-induced electromotive force of the Galfenol cantilever beam can be established under the excitation force of the energy harvester, with the model parameters shown in Table 2.

By solving the mathematical model of the induced electromotive force with different mass blocks of 2 g, 4 g, and 8 g, respectively, the induced electromotive force converted by a single Galfenol cantilever beam in the wave energy harvester can be obtained, as shown in Figure 9.

As can be seen from Figure 9, under a certain bias magnetic field, the magnitude of the induced electromotive force of the Galfenol cantilever beam is approximately sinusoidal, which is mainly due to the fact that the deformation of the Galfenol cantilever beam has a sinusoidal trend. Similar to the law of the amplitude change in the Galfenol cantilever beam’s deformation displacement caused by the change in mass blocks, the magnitude of the induced electromotive force of the Galfenol cantilever beam slightly increases with the mass of the mass block. Among them, the induced electromotive force curve of the 2 g mass block has a smaller peak-to-peak change and is relatively stable, with an induced electromotive force amplitude of 8.96 mV.

## 4. Experimental Verification

In order to verify the effectiveness of the transient dynamic simulation results of the Galfenol cantilever beam and the mathematical model of the wave energy harvester, an experimental platform was built for verification. Due to the special nature of the marine environment and the lack of a wave-making water tank experimental platform, it is difficult to build an experimental platform that is the same as the actual wave condition environment. Therefore, an approximate simulation method was used to establish an experimental platform for verifying the energy harvester. Based on the sine excitation signal of the dynamic simulation of the Galfenol cantilever beam structure, a sinusoidal motion oscillator was designed to simulate the vibration displacement of the energy harvester under wave loads. The working principle schematic diagram of the experimental platform is shown in Figure 10. The T-shaped rod of the sinusoidal motion oscillator is driven by the motor through the PLC and servo driver to perform the sinusoidal motion. The Galfenol cantilever beam fixed on the T-shaped rod undergoes deformation under the excitation of sinusoidal motion. Through the coupling of the magnetostrictive inverse effect and the electromagnetic induction law, electric energy is generated in the coil wound around the cantilever beam.

In order to detect the electrical energy output capability of the energy harvester, a strain gauge was attached to the root of the Galfenol cantilever beam. Then, it was connected in sequence with the Wheatstone bridge circuit, data acquisition card, and computer. The corresponding voltage change graph was obtained through the testing program in LabVIEW2014 software. The induction coil was connected to an oscilloscope to observe the change in induced electromotive force. The experimental apparatus platform constructed is shown in Figure 11.

### 4.1. Analysis of Experimental Verification Results for the Transient Dynamic Simulation

The voltage value measured by the strain sensor was converted into the corresponding deformation value of the Galfenol cantilever beam using the calculation formula of the Wheatstone bridge. Then, the contrast graph between the theoretical deformation value and the experimental deformation value of the Galfenol cantilever beam was drawn, as shown in Figure 12.

According to the conversion principle of the strain gauge, the change in the strain gauge voltage signal is proportional to its own deformation, and the deformation amount of the strain gauge is basically consistent with that of the Galfenol cantilever beam. The reason for the waveform asymmetry and linear distortion here is that the cantilever beam bends downward due to the action of the mass block, and the end of the cantilever beam does not symmetrically move up and down when the energy harvesting device floats up and down with the wave. When the energy harvesting device moves upward with the wave, the movement of the root of the cantilever beam is opposite to the direction of the force on the mass block at the end, and the deformation of the cantilever beam increases. When the energy harvesting device moves downward with the wave, the movement of the root of the cantilever beam is in the same direction as the force on the end mass, and the deformation of the cantilever beam decreases. As shown in the comparison of the theoretical value and the experimental value of the strain-induced voltage in Figure 12, the theoretical value and the experimental value are basically consistent, which proves that the simulation results of the self-deformation of the Galfenol cantilever beam obtained during the transient dynamic simulation process are accurate and consistent with its deformation law. The error between the two is maintained within 7%, thereby demonstrating the accuracy of the transient dynamic simulation results.

### 4.2. Analysis of Experimental Validation of the Wave Energy Harvester Mathematical Model

In order to conduct a more thorough comparison and analysis of the experimental and theoretical induction voltage data for the wave energy harvester under sinusoidal excitation, the data points obtained from the experiment were plotted together with the theoretical data points on the same graph, as shown in Figure 13. The root-mean-square error (RMSE) between the experimental and theoretical induction voltage curves was then calculated, and the results are shown in Table 3.

When a resistance of 20 ohms is chosen as the load, substituting the experimental value of the induced voltage, the actual output power can be obtained, as shown in Figure 14.

Analysis of the comparison between the theoretical and experimental voltage values in the above Table 3 shows that the induced voltage generated by the energy harvesting mechanism exhibits sinusoidal variation under excitation by a sinusoidal signal similar to a wave motion, and the waveforms of the measured induced voltage in the experiment and simulated induced voltage from the mathematical model of the device are basically consistent. The maximum induced voltage value obtained through the mathematical model simulation is 8.96 mV, while the maximum measured induced voltage value is 7.51 mV, and the Maximum output power is 3.22 μW. The main reason for the discrepancy between the theoretical and experimental values is the magnetic leakage phenomenon in the Galfenol sheet, and the influence of environmental noise on the oscilloscope measurement in the experiment. Calculation results indicate that the error between theoretical and experimental values is mainly distributed around 15%, thereby verifying the ability of the mathematical model of the wave energy harvester to accurately reflect the variation trend in voltage over time. An analogous calculation based on the experimental results shows that the energy harvester produces an induced voltage of 30.04 mV and an output power of 12.88 μW if a maximum error of 17% between experiment and theory is taken into account.

## 5. Conclusions

(1)In order to solve the problem of energy supply difficulty and high cost in marine wireless sensor networks, a wave energy harvester with a cylindrical, multi-cantilever beam symmetrically distributed structure was designed using the magnetostrictive material Galfenol sheet. The energy harvester is designed to float on the water’s surface without capsizing, utilizing a float board and gravity anchor. This design ensures the device’s stability in harsh ocean environments and maximizes energy harvesting efficiency.(2)The dynamic equation of the device was established to determine the motion displacement of the device under wave excitation. The transient dynamic simulation analysis module in ANSYS Workbench was used to simulate and analyze the motion characteristics of the Galfenol cantilever beam, and to determine its deformation law. Based on the Jiles-Atherton hysteresis theory model, the mathematical model of the strain displacement and induced electromotive force of the Galfenol cantilever beam was established.(3)The error between the transient dynamic simulation results and the experimental results of the cantilever beam was kept within 7%. The voltage that could be generated by the energy harvester was 30.04 mV, and the power output was 12.88 μW. The error between the theoretical and experimental values of the energy conversion mathematical model was about 15%, which verifies the validity of the device mathematical model.

This study utilized linear wave theory to analyze the forces and conduct dynamic simulations of the energy harvester. However, there may be discrepancies between the simulated environment and real wave conditions. In the future, a more accurate wave equation can be established to improve the accuracy of the device’s output results through force simulation analysis. Additionally, the device will be tested and analyzed on the actual sea surface to validate the results of the study.

## Figures and Tables

**Figure 1 materials-16-05585-f001:**
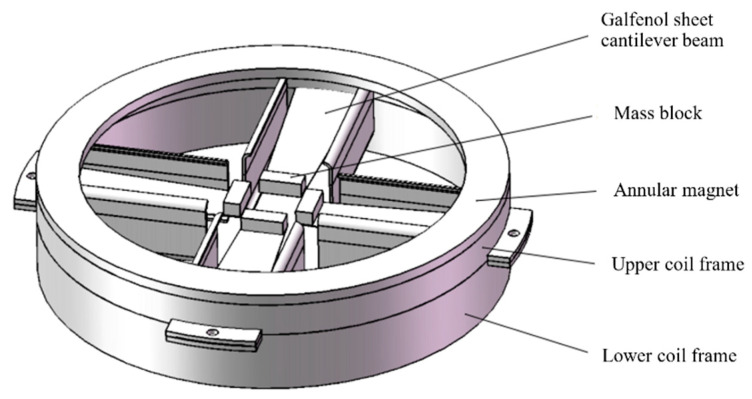
Ring-shaped energy harvesting mechanism.

**Figure 2 materials-16-05585-f002:**
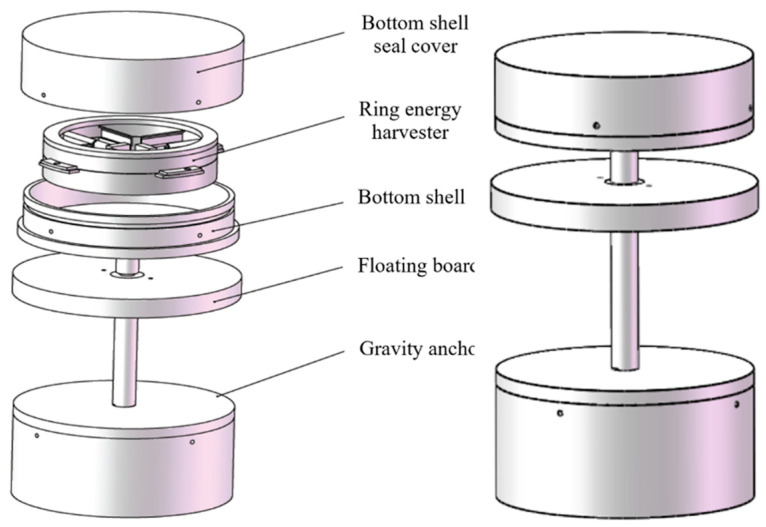
Schematic diagram of the 3D model of the wave energy harvester.

**Figure 3 materials-16-05585-f003:**
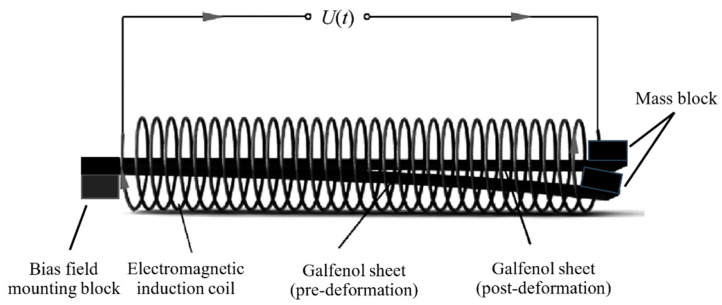
Working schematic diagram of Galfenol energy harvester.

**Figure 4 materials-16-05585-f004:**
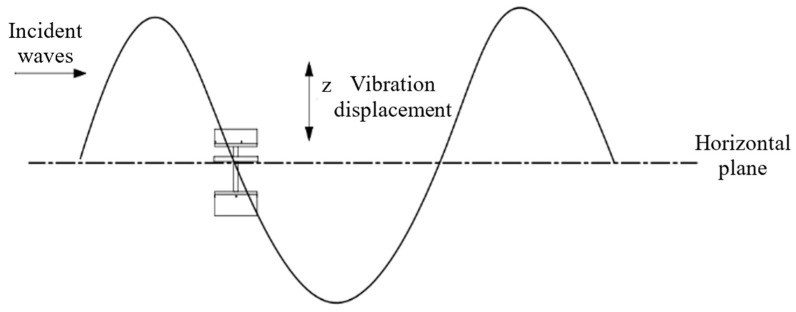
Schematic diagram of the movement of the wave energy harvester.

**Figure 5 materials-16-05585-f005:**
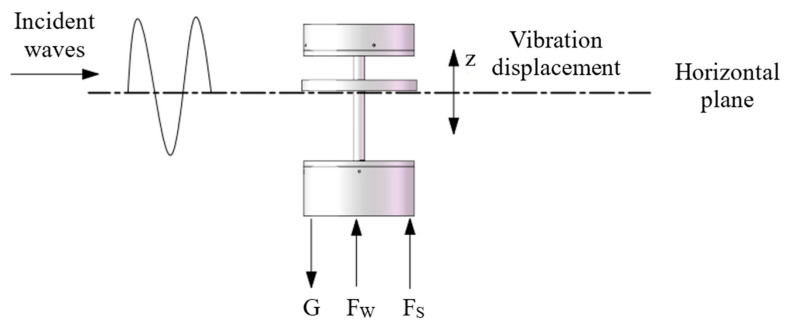
Schematic diagram of the force of the wave energy harvester.

**Figure 6 materials-16-05585-f006:**
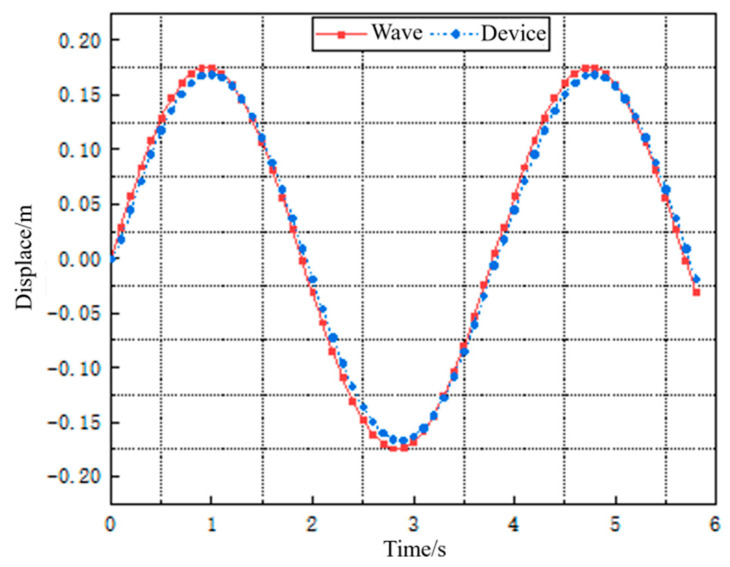
Displacement curve of wave energy harvester.

**Figure 7 materials-16-05585-f007:**
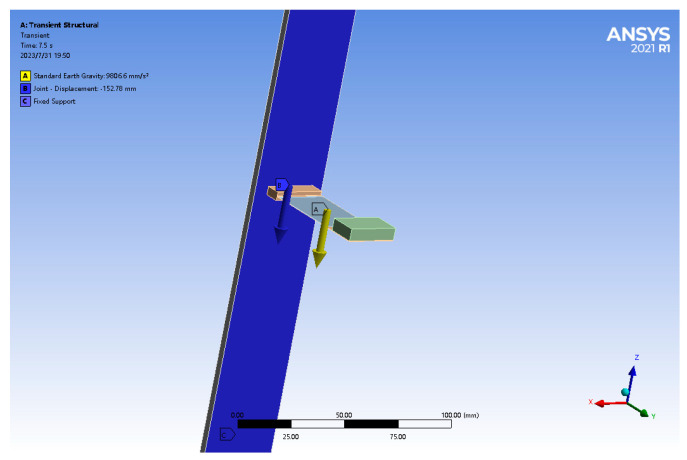
Schematic diagram of Galfenol cantilever beam restraint, gravity, and load settings.

**Figure 8 materials-16-05585-f008:**
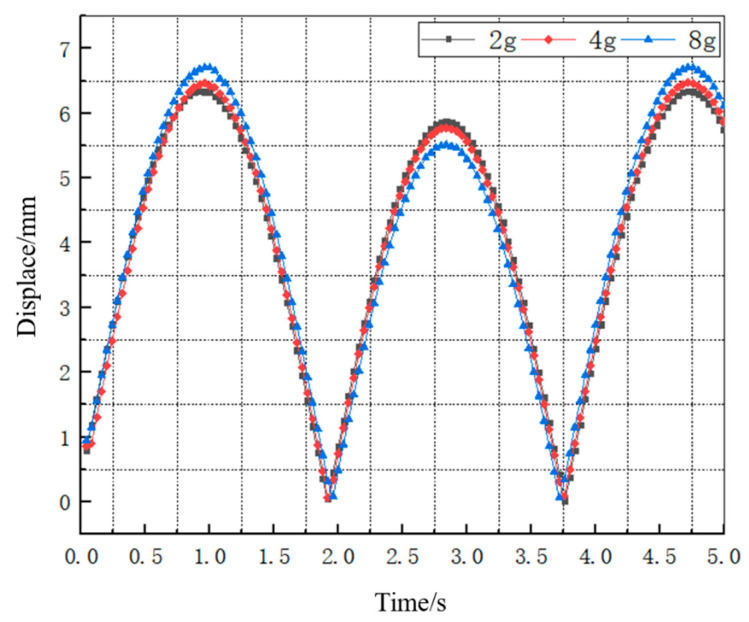
Self-deformation curve of Galfenol cantilever beam under 2 g, 4 g, and 8 g mass blocks.

**Figure 9 materials-16-05585-f009:**
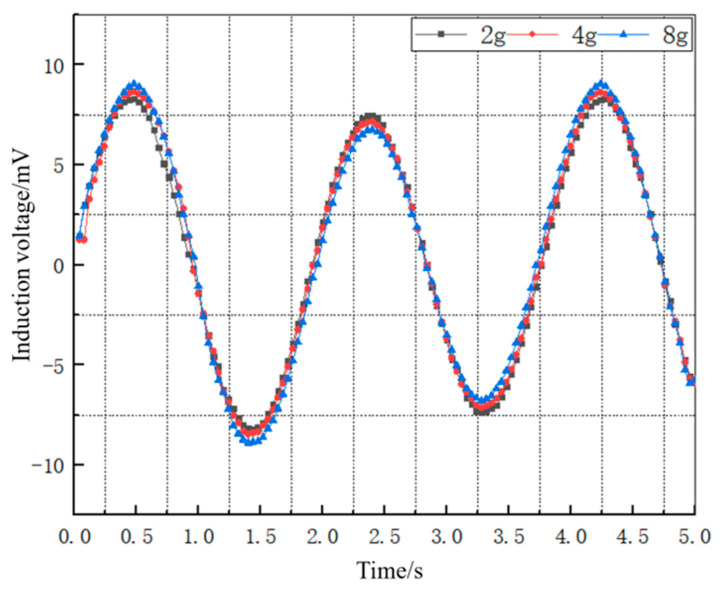
Induced electromotive force of Galfenol cantilever beam with different masses.

**Figure 10 materials-16-05585-f010:**
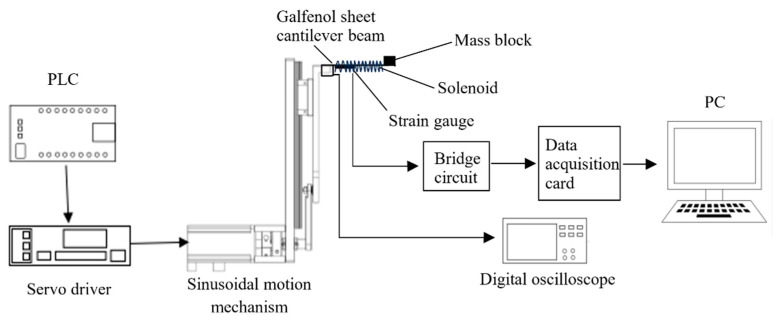
Schematic diagram of the experimental principle.

**Figure 11 materials-16-05585-f011:**
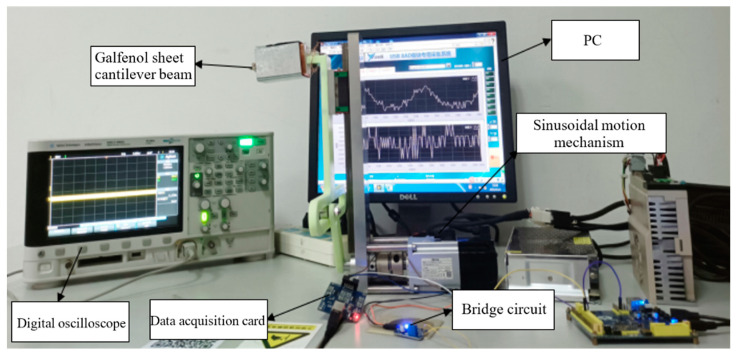
Experimental device platform.

**Figure 12 materials-16-05585-f012:**
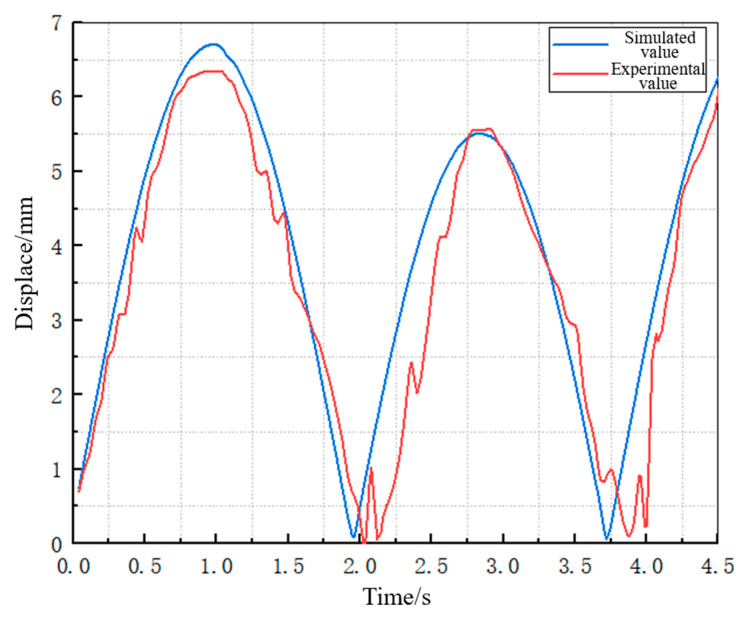
Theoretical and experimental values of the deformation of Galfenol sheet.

**Figure 13 materials-16-05585-f013:**
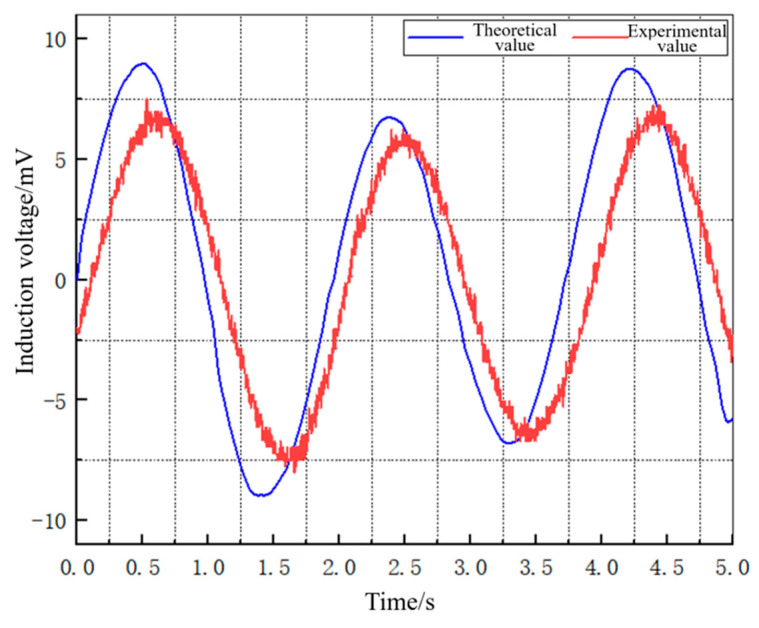
Comparison of experimental and theoretical induced voltage curves.

**Figure 14 materials-16-05585-f014:**
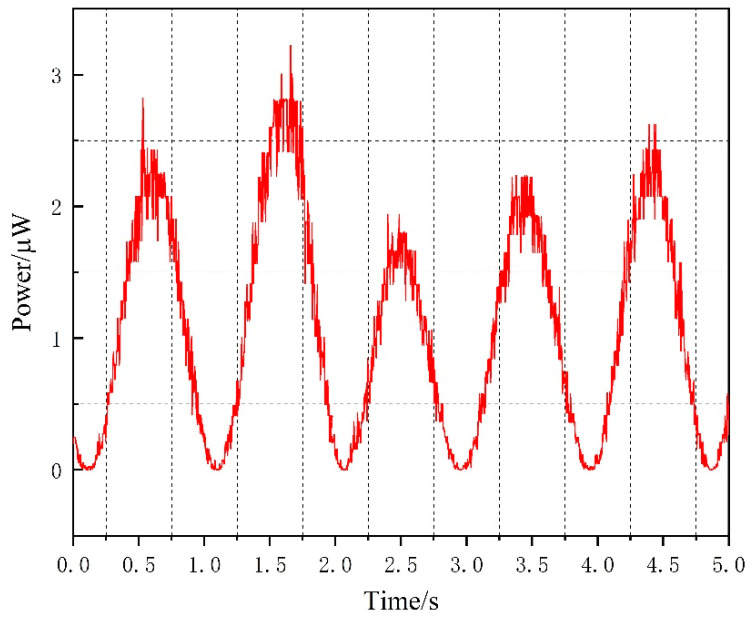
Actual output power.

**Table 1 materials-16-05585-t001:** Elements of wave conditions.

Wave Height (m)	Periodicity (s)	Frequency (rad/s)	Wave Length (m)	Wave Number (k)
H	T	$\omega =\frac{2\pi }{T}$	$\lambda =\frac{gT^{2}}{2\pi }$	$K=\frac{2\pi }{\lambda }$
0.349	3.784	1.66	22.3	0.28

**Table 2 materials-16-05585-t002:** Theoretical model parameter settings.

Model Parameters	Numerical Values
Saturation magnetization strength $M_{s}$/(A/m)	$2.71\times 10^{4}$
Hysteresis-free magnetization intensity factor a	7012
Constant c	0.18
Elastic modulus E/(pa)	$3\times 10^{10}$
Vacuum magnetic permeability $\mu_{0}$	$4\pi \times 10^{-7}$
Number of turns N	700
Cross-sectional area A/(m)	$1.25\times 10^{-4}$
Galfenol length l/(m)	0.05
Compressed magnetic stress constants e	166
Magnetic fields caused by Prestressing $H_{\sigma }$/(A/m)	$2.6\times 10^{8}$
Saturated hysteresis $\lambda_{s}$ (ppm)	160

**Table 3 materials-16-05585-t003:** Error analysis of experimental and theoretical induced voltage curves.

Peak-Trough Sequence	Theoretical Values (mV)	Experimental Values (mV)	Absolute Errors (mV)	Errors
1	8.96	7.51	−1.45	16%
2	−8.93	−7.75	1.18	13%
3	6.73	6.23	−0.5	7.4%
4	−6.88	−6.36	0.52	7.5%
5	8.75	7.24	−1.52	17%

## Data Availability

Data underlying the results presented in this paper are not publicly available at this time but may be obtained from the authors upon reasonable request.

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
