# Peer review of "Modeling and Analysis of Wave Energy Harvester with Symmetrically Distributed Galfenol Cantilever Beams"

_materials, 2023, doi:10.3390/ma16165585_

Round 1

Reviewer 1 Report

The paper describes the modeling analysis of an innovative wave energy harvester designed by authors. The device in question is based on a magnetostrictive material, i.e. Galfenol. The authors provide a mathematical model of the device together with FEM simulations and an experimental validation of simulated results.

The subject of the paper seems very interting and the paper itself is generally well-written.

Despite the reviewer appreciated a lot the efforts made by the authors especially to set the experimental environment and tests, I am strongly convinced that something could and then needs to be improved in order to provide even more value to the paper.

Therefore, I suggest the Editor to accept the paper after a MAJOR REVISION.

Please, find in the following a detailed description of the reviewer’s concerns and some specific remark on the text.

The model provided by authors for the whole device is too simplified, and then possibly wrong, for several reasons.

1) Main assumption: the device moves only vertically. This is not true, but somehow this could be still accepted, especially under some assumptions that however need to be clearly stated by authors. 

2) The FEM modeling of the whole device (composed by four cantilever beams) is reduced to just one cantilever beam (with maybe a mass at the end: this is not clear). A reliable model of the device, however, could be obtained by simply considering a periodic boundary condition in the FEM model. This allows to have a clear picture of the whole device's behavior with a reduced computational burden. Moreover, no details are given on the main quantities used by authors for modeling the Galfenol, e.g. mass density, thickness of the Galfenol layer, etc. Then, from Figure 6 it is completely impossible to understand also what has been modeled and how.

----- Abstract -----

1) Please, specify within the abstract acronyms like "J-A" (not necessarily all the scholars could be familiar with hysteresis models);

2) it is not clear to me what authors means with "deformation displacement": we normally describe the effect of a stress analysis chooding as variable either deformation (adimensional) or displacement.

3) last sentence seems to me too strong since the model error is around 15 % (not exactly negligible...). Therefore, I would rather describe the analysis as a preliminary one that could be even improved.

% ---- Introduction ----

row 43) please, avoid degrees like "academician", etc.

row 47) it was impossible to me to fine the referenced paper by Liu. Moreover, I guess that the energy conversion is not a direct one, but the Faraday-Neumann effect should be exploited as well.

row 54) what do authors mean with "ultra-magnetostrictive materials"? This definition, differently from the "giant magnetostrctive materials one", is not spread (as far as the reviewer knows) within the magnetostrictive community.

% ----- Section 2 -----

title) why not all the words have a capital letter? Please, avoid capital letters if unnecessary. (this remark could be applied also to following sections)

rows 73-78) Please, specify also that for efficiency reasons, cantilever beams need to work close to resonance frequency.

Figure 1) few words are truncated in the figure (e.g., "Annular magn", etc.)

row 90) "the largest fluctuations": compared with? is it an absolute estimation?

row 91) "to the energy harvester to harvest wave energy": redundant

rows 102-104) please, mention Faraday-Neumann's law

% ---- Section 3 ----

row 111) "the device can be approximated as a simple harmonic motion": the subject is wrong.

row 128) "z is the displacement of the wave in the vertical direction": this is alittle bit confusing. In figures 3 and 4, "y" is the the vertical axis 

row 153) "the characteristic dimensions of the structure": what is that? The diameter?

row 214) mu_0 is the magnetic permeability of the vacuum? If so, this is wrong.

row 213 and 218) equation 7) and 8) are not in agreement. If the 8) is correct, then the 7) should be: B = mu_0*(H+M). What is the unit of measure of "M"?

% ---- Section 4 ----

row 276) What do you mean with "sine mechanism"?

row 284 to 290) are the authors analyzing both the signals coming from the strain gauges and the voltage coming from the induction coil? Moreover, the measurement on the voltage across the induction coil has been performed in open-circuit?

The paper is generally well-written.

Reviewer 2 Report

This research is very well related to the future needs. However, it suffers from the following points.

1-      A comprehensive schematic is required to explain the principle of the harvester.

2-      What is the effect of external loads on the voltage and power?

I think this recent two papers can be used as the reference for Galfenol and harvester.

·        ‘’Novel Contactless Hybrid Static Magnetostrictive Force-Torque (CHSMFT) Sensor Using Galfenol’’ Journal of Magnetism and Magnetic Materials, Vol.553, p. 168969 (2022).

·        "Modeling and Characterization of Permendur Cantilever Beam for Energy Harvesting," Energy, vol. 176, pp. 561-569, (2019).

Needs improvement!

Reviewer 3 Report

minor corrections. Can be easily introdued upon reading through by native speaker

Round 2

Reviewer 1 Report

All the suggestions provided by the reviewer have been correctly implemented.

Author Response

Thanks again for the revisions!

Reviewer 3 Report

The revised version of this paper is much easier to read and can thus be accepted for publication.

Before publication line 145 should be corrected. In this line the vertical displacement is still incorrectly denoted by y. Please change y --> z.

As a suggestion from my side for future work: please measure the restoring force of your four-cantilever arrangement as a function of displacement in z-direction. I guess that you will find that the restoring force will increase with z. In case this proves to be correct, the spring constant k in Equation z will become eleongation-dependent and then produce non-linearity in the vibration  behavior; in particular a much changed resonance behavior. Maybe this will not be relevant as wave motion frequencies are much lower than the resonance frequencies of the vibrator system.  

Author Response

The displacement variable in line 145 of the text has been changed and and we thank you again for your revisions to the manuscript and suggestions for future work.